# NODEATTACK: ADVERSARIAL ATTACK ON THE ENERGY CONSUMPTION OF NEURAL ODES

## ABSTRACT

Recently, Neural ODE (Ordinary Differential Equation) models have been proposed, which use ordinary differential equation solving to predict the output of neural network. Due to the low memory usage, Neural ODE models can be considered as an alternative that can be deployed in resource-constrained devices (e.g., IoT devices, mobile devices). However, to deploy a Deep Learning model in resource-constrained devices, low inference energy cost is also required along with low memory cost. Unlike the memory cost, the energy consumption of the Neural ODEs during inference can be adaptive because of the adaptive nature of the ODE solvers. Attackers can leverage the adaptive behaviour of Neural ODEs to attack the energy consumption of Neural ODEs. However, energy-based attack scenarios have not been explored against Neural ODEs. To show the vulnerability of Neural ODEs against adversarial energy-based attack, we propose `NODEAttack`. The objective of `NODEAttack` is to generate adversarial inputs that require more ODE solvers computations, therefore increasing neural ODEs inference-time energy consumption. Our extensive evaluation on two datasets and two popular ODE solvers show that the samples generated through `NODEAttack` can increase up to 168% energy consumption than average energy consumption of benign test data during inference time. Our evaluation also shows the attack transferability is feasible across solvers and architectures. Also, we perform a case study showing the impact of the generated adversarial examples, which shows that `NODEAttack` generated adversarial examples can decrease 50% efficiency of an object-recognition-based mobile application.

## 1 INTRODUCTION

Deep Neural Networks (DNNs) have shown great potential in many challenging tasks (image classification, natural language process, and playing games). To cope with tasks with higher complexity, the number of DNN parameters is increasing rapidly. Because of this reason, DNNs require considerable memory usage both in training and inference. To address the issue of increased memory usage, researchers simulate the solver of ordinary differential equation (ODE) and propose Neural ODE techniques (Chen et al., 2018). Neural ODE does not store any intermediate quantities of the forward pass and allows us to train DNNs with constant memory cost. Neural ODE also performs better than traditional DNNs for irregularly sampled time series data. Because of decreased memory cost, Neural ODEs are viable options to be used in resource-constrained devices like mobile devices or UAVs (Unmanned Aerial Vehicle).

Due to the adaptive energy consumption of Neural ODE models (Section 3), the model robustness in terms of energy consumption or energy robustness (Defined in 4.1) of the model needs to be investigated to deploy Neural ODE models in resource-constrained devices. Otherwise, lack of energy robustness in Neural ODEs can lead to tragic situations. For example, we assume that Neural ODE model is deployed for mobile apps, which are used to help visually impaired people. The energy consumption of the model is not robust, then the battery of the mobile device will be drained faster. This can be fatal for the visually impaired person. To avoid such scenarios, evaluating energy robustness of Neural ODEs is required to avoid unwanted incidents.

Although energy robustness of Neural ODEs has not been explored, unlike accuracy-based robustness. Recent work shows that Neural ODE models are more robust against accuracy-based adversarial attacks (Yan et al. (2019)) than traditional DNNs. However, finding a relationship between input and energy consumption is more challenging because the relation between input and energy consumption of DNNs is not well-defined. To explore the energy robustness of Neural ODEs, the relationship between input and energy consumption of Neural ODEs needs to be defined.

Recent works, ILFO (Haque et al. (2020)) and DeepSloth (Hong et al. (2020)), have evaluated energy robustness of Adaptive Neural Networks (AdNNs) by proposing white-box attack. However, optimizing the loss function proposed by the aforementioned AdNN attacks can not evaluate the energy robustness of Neural ODEs. AdNNs activate or deactivate certain components of DNNs based on the intermediate outputs of certain computing units and consumes a different amount of energy based on different inputs. Both the attack's objective is to increase the number of activated DNN components by modifying the specific computing unit outputs, and both attack use specific loss function optimization to achieve that. However, Neural ODE functionality is different than traditional AdNN functionality because, for Neural ODE, no component is deactivated or activated during inference. The adaptive behavior of a Neural ODE model depends on the adaptive ODE solver used to predict the output. Furthermore, for a specific trained Neural ODE model, we can find variable energy consumption for single input depending on the type of ODE solver used, where for traditional AdNNs, energy consumption based on a specific input will be the same always for a specific trained AdNN. Therefore, a novel approach is needed to explore the energy robustness of Neural ODEs.

To explore the energy robustness of Neural ODEs, We propose `NODEAttack`, a white-box approach that uses step-size of the ODE solvers to formulate attack. ODE solvers use an iterative way to approximate a function, and the objective of our approach is to increase the number of iterations, increasing the energy consumption of Neural ODEs. Our attack formulation is based on the fact that decreasing step-size would increase the number of iterations. Specifically, we develop two attack techniques to evaluate Neural ODE's energy robustness, namely Input-based attack and Universal attack. Input-based attack evaluates energy robustness where testing inputs are semantically meaningful to the Neural ODE model (*e.g.,* meaningful images). On the other hand, Universal attack evaluates worst-case energy robustness where each testing input maximizes the energy consumption for each target ODE solver. To the best of our knowledge, this is the first energy-based adversarial attack against Neural ODEs.

We evaluate `NODEAttack` on mainly two criteria: effectiveness and transferability using the CIFAR-10 and MNIST datasets (Krizhevsky et al. (2009); Deng (2012)). We evaluated `NODEAttack` on two popular ODE solvers: Dopri5 (Dormand and Prince (1980)) and Adaptive Heun Süli and Mayers (2003). We evaluate the the effectiveness of `NODEAttack` against natural perturbations and corruptions Hendrycks and Dietterich (2019a). We observed that `NODEAttack` generated adversarial inputs can increase up to 168 % energy consumption than the average energy consumed by benign test inputs. Also, we noticed that transferability is feasible between two Neural ODEs differentiated by ODE solver or model architecture.

Our paper makes the following contributions:

- **Problem Formulation and Approach.** Our work is the first attempt to formulate energy-based adversarial attack against Neural ODE models. Also, our work proposes a novel loss function based on step-size of ODE solvers to generate adversarial inputs.

- **Evaluation.** We evaluate our approach across two ODE solvers and two datasets based on two criteria.

## 2 BACKGROUND AND RELATED WORKS

### 2.1 NEURAL ORDINARY DIFFERENTIAL EQUATIONS

Neural Ordinary Differential Equations (Neural ODE) (Chen et al. (2018)) have been successful in attaining accuracy close to the State of the Art DNN techniques but with lesser memory consumption. Neural ODEs incorporate Ordinary Differential Equations solvers into Neural Network

architectures. Models such as residual networks and recurrent neural network decoders create complicated transformations by devising a sequence of transformations to a hidden state:

$$h_{t+1} = h_t + f(h_t, \theta_t)$$

Operation of a residual block can be interpreted as the discrete approximation of an ODE where the discretization step value is one. In a neural ODE, the discretization step is set to zero and the relation between input, and output is characterized by the following set of equations:

$$\frac{dh(t)}{dt} = f(h(t), t, \theta), h(0) = h_{in}, h_{out} = h(T)$$

Solving $h(T)$ gives the output and ODE solvers can be used for that purpose.

Additionally, Proposed work by Quaglino et al. (2019) expresses the Neural ODE dynamics as truncated series of Legendre polynomials and accelerate the model. Dupont et al. (2019) explores the limitations in approximation capabilities of neural ODEs because of the preserving of input topology. Recent work by Yan et al. (2019) explore the robustness of Neural ODEs against Neural ODEs and propose TisODE to increase the robustness of Neural ODEs. However, no other work has focused on the energy robustness perspective or Neural ODEs, and to our knowledge, this is the first work in that direction.

## 2.2 Runge Kutta Method

Runge Kutta method (Runge (1895); Kutta (1901)) is an ODE solver which solved ordinary differential equations through approximation. First-order differential equation given by,

$$\frac{dy(t)}{dt} = y^{'}(t) = f(y(t), t)$$

, with $y(t_0) = y_0$ Here $y$ is an function of time $t$ and $y^*$ is the value of $y$ at $t = 0$. Four slope approximations $k_1, k_2, k_3, k_4$ are used to estimate approximate value of $y$ ($y^*$) at $t = t_0$ (Detailed equation in Appendix).

Final estimate of $y^*(t_0 + h)$ can be represented as,

$$y^*(t_0 + h) = y^*(t_0) + (\frac{1}{6}.k_1 + \frac{1}{3}.k_2 + \frac{1}{3}.k_3 + \frac{1}{6}.k_4).h$$

Here, $h$ is the step size. This is called fourth order Runge Kutta Method, because the local error (approximation error at a particular time ) for step-size h is $O(h^4)$. For better approximation of function, multiple works (Dormand and Prince (1980); Süli and Mayers (2003)) have proposed to use adaptive step size.

## 2.3 Adversarial Examples

Adversarial Examples are the inputs that are fed to machine learning models to change the prediction of the model. In earlier works by Dalvi et al. (2004); Lowd and Meek (2005); D.Lowd and C.Meek (2005), 'good word attacks' or spelling modifications have long been used to bypass the spam filters. More recently, Szegedy et al. (2013) and Goodfellow et al. (2014) propose adversarial attacks on deep computer vision models. Karmon et al. (2018) propose a technique to attack CNNs in which a localized patch is introduced in an image instead of adding noise to the full image. With a similar approach, adversarial attacks have been extended to various fields like text and speech processing (Carlini et al. (2016); Jia and Liang (2017)), and graph models (Zügner et al. (2018); Bojchevski and Günnemann (2019)). Recently, Haque et al. (2020); Hong et al. (2020) have proposed adversarial energy based attacks against Adaptive Neural Networks. However, as mentioned in the introduction, existing attacks can not be used to increase energy consumption of Neural ODEs.

## 3 Adaptive Nature of Neural ODE

In this section, we discuss the reason of adaptive energy consumption of Neural ODEs. As discussed in Section 2, ODE solvers use iterative approximation to calculate the function value at a certain

point. If the number of iterations is increased, the function $f$ is needed to be calculated for a higher number of times, increasing the energy consumption of the process. The number of iterations can be increased by decreasing the step-size, which we discuss in Section 4.2.

We also have investigated the adaptive nature of Neural ODEs through a preliminary study. We have trained a Neural ODE CNN model (Chen et al. (2018)) with MNIST (Deng (2012)) training data and used the MNIST test data for inference. For ODE solver, we used dopri5 (Dormand and Prince (1980)) ODE solver. For each test data, we measure energy consumption using Nvidia TX2 server.

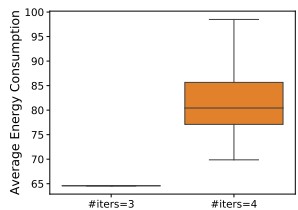

From the results, we have noticed that ODE solver takes a different number of iterations (3 and 4 in this experiment) to approximate a function. Figure 1 describes our findings. When the number of iterations is 3, average energy consumption drops more than 15J. However, the range of inference energy consumption is limited is low for benign in-distribution data.

Figure 1: Difference in energy consumption with different number of iterations,

## 4 NODEATTACK

We elaborate the approach of NODEAttack through this section. First, we define two types of energy robustness in this section. Based on the definitions, we formulate the problem and propose two type of energy attacks against Neural ODEs.

### 4.1 ENERGY ROBUSTNESS

We define energy robustness for Neural ODEs in two ways: Input-based Energy Robustness ($E_i$) and Universal Energy Robustness ($E_u$). $E_i$ is defined based on the maximum energy consumed by the model for an input which belongs to the training data distribution of the model. Let us assume, $x$ is an input that is within the data distribution of a DNN $f$. We want to add perturbation $\delta$ to $x$ such that energy consumption is maximum. In that scenario, $E_i$ can be represented as,

$$E_i = -\max_{\delta \in R} ENG_f(x + \delta)$$

, where $R$ is set of admissible perturbations such that $x + \delta$ remains within distribution, and $ENG_f$ represents the energy consumption of DNN $f$.

$E_u$ can be described based on the highest possible energy consumed by a model for any input. Inputs used to measure $E_u$ can be out-of-distribution inputs also. For a DNN $f$ and any input $x$, $E_u$ can be represented as,

$$E_u = -\max_x ENG_f(x)$$

, where $ENG_f$ represents energy consumption of DNN. As $E_i$ and $E_u$ represents the highest energy consumption in two different cases, if we increase the the value of $E_i$ and $E_u$, the maximum energy consumption of the model would decrease. As our objective is to decrease model's maximum energy consumption, by increasing the value of $E_i$ and $E_u$, energy robustness of a model can be increased.

### 4.2 PROBLEM FORMULATION

The objective of our approach is to create adversarial samples which can increase energy consumption of Neural ODEs. For this purpose, we mainly focus on increasing the number of iterations needed for approximation in ODE solvers. As the energy consumption to approximate a value at any point using ODEs does not vary significantly across all the points, increasing the number of iteration would increase the energy consumption in Neural ODEs.

The number of iteration in an ODE solver is dependent on the step size. Each ODE solver approximates the function based on differentiation of the function at different instance and step-size. ODE solvers modify the step-size based on the function slope. If the slope is high, then a smaller step-size is required for the approximation. For smaller step-size, the solver would use more iteration for approximation causing higher energy and time consumption. However, if the slope is low, larger step size can be used for approximation.

For ODE solvers, step-size is modified based on difference in approximations when the function is approximated multiple times at time t. The multiple approximations can be performed with different step-size, or with using a higher order and a lower order approximation (Press and Teukolsky (1992)). For example, Runge-Kutta-Fehlberg (Fehlberg (1969)) Method creates two different Runge Kutta method with different orders (order four and five), but with same intermediate values [1]. The fifth order approximation at time $t = t_0 + h$ can be represented as,

$$y^5(t_0 + h) = y^5(t_0) + c1.k_1 + c2.k_2 + c3.k_3 + c4.k_4 + c5.k_5 + c6.k_6$$

and, the fourth order approximation at time $t = t_0 + h$ is,

$$y^4(t_0 + h) = y^4(t_0) + d1.k_1 + d2.k_2 + d3.k_3 + d4.k_4 + d5.k_5 + d6.k_6$$

Here, $k_1, k_2, k_3, k_4, k_5, k_6$ are the intermediate values, $c1, c2, c3, c4, c5, c6, d1, d2, d3, d4, d5, d6$ are the constants. The difference between both approximations can be defined as error estimate,

$$\Delta \equiv y^5 - y^4$$

. Based on the value of $\Delta$ at a certain t, next step-size is calculated. If the error value $\Delta$ is greater than a certain desired error value, then the step size is decreased than the previously used step size, on the other hand, if $\Delta$ is lesser than the desired error value, the step size is increased. Therefore, it can be concluded that the adaptive step size is dependent on $\Delta$.

As the adaptive step size is dependent on $\Delta$, and the $\Delta$ is dependent on multiple approximations of the function, we can refer that the adaptive-step-size is dependent of the initial value of t ($t_0$). Therefore, for a Neural ODE, the adaptive-step-size and the number of iterations to approximate the function will be a function of the model input. We assume that a solver takes $N(X)$ iterations to approximate a function for input $X$, where $h_1(X), h_2(X), ..h_N(X)$ is the step-size used for N iterations. We define the average step-size as,

$$h_{avg}(X) = \frac{h_1(X) + h_2(X) + .. + h_N(X)}{N(X)}$$

We have mainly two objectives. Our first goal is to find the input $X$ for which $h_{avg}(X)$ is minimum to evaluate the Universal Energy Robustness ($E_u$) of Neural ODEs.

$$X_{adv} = \operatorname*{argmin}_X h_{avg}(X)$$

Our second objective is to evaluate Input-based Energy Robustness of Neural-ODEs. To address that, we find the perturbation $\delta$, which can be added to an in distribution input $X$ such that $h_{avg}(X + \delta)$ is minimum.

$$\delta_{adv} = \operatorname*{argmin}_{\delta \in R} h_{avg}(X + \delta)$$

Here, $R$ is set of admissible perturbations such that $x + \delta$ remains within distribution

## 4.3 APPROACH

For exploring the energy robustness of Neural ODEs, we use a gradient-based optimization technique to create adversarial inputs. Based on the objectives defined in Section 4.2, we propose two strategies to evaluate Neural ODE: Input-based attack and Universal attack.

**Input-based attack.** In this approach, we create adversarial inputs for each input image such that the semantic meaning of the input is preserved. We assume that $x$ is a test input. Our objective is to find perturbation $\delta$, adding which the energy consumption of the model increases. We can define the problem as,

$$minimize(|\delta| + c \cdot f(x + \delta)) \text{ such that, } (x + \delta) \in [0, 1]^n \tag{1}$$

where $f(x + \delta) = h_{avg}(x + \delta)$ and $c$ is a predefined constant. If $c$ value increases, the average step size would have greater weightage in the optimization problem, leading to a more noisy input.

The algorithm is explained through Figure 2 and Algorithm 1. The algorithm can be divided into following parts. *Initializing Perturbation* is the first step where we initialize different variables, including perturbation, which are going to be modified by the optimization. All the next steps are

---

[1]For the equation mentioned in Section 2.2, k1,..k4 are the intermediate values

performed iteratively. In the second step (*Get Step Size*), the average step size of the ODE solver is calculated based on the modified input generated after adding the initialized perturbation to input image. Next, we calculate the loss function value based on average step size and optimize the function (*Optimizing Loss Function*). Then, we denormalize and normalize the modified input to nullify the effect of information loss due to denormalization (*Denormalizing and Normalizing Adversarial Input*). Next, we consider the perturbation added to denormalized input as updated perturbation and we calculate the number of steps induced by denormalized input. If the current number of steps is greater than previously recorded number of steps, we store the current input perturbation as most successful perturbation (*Update Perturbation and Record Most Successful Perturbation*). All the aforementioned steps are explained below.

**Initializing Perturbation**: This step is used for initializing variables. We initialize perturbation ($\delta$), maximum recorded iteration of ODE solver($Max\_N$), and highest energy-consuming adversarial input ($X\_best$) (Line 2). Number of iterations ($T$) is also initialized (Line 3).

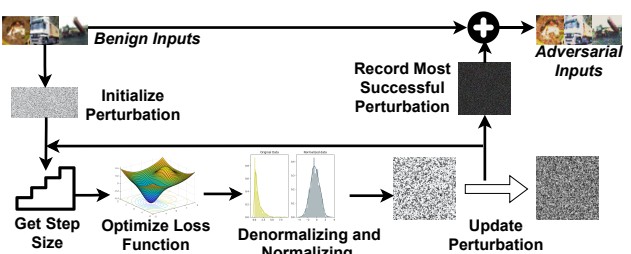

Figure 2: Overview of Input-based Attack

**Get Step Size**: In this step, a perturbed image ($X'$) is generated by adding initialized perturbation ($\delta$) to the input image ($X$) (Line 6). Then, $scale$ method re-scales the $X'$ using $tanh$ function. Next, average step size of the solver for $X'$ is received using $avg\_step\_size$ method (Line 7).

**Optimizing Loss Function**: In this step, the loss function value is calculated first. The loss function $L$ consists of two components: first, the euclidean distance between the input image ($X$) and perturbed image ($X'$) (calculated on Line 8), and the average step size ($h_{avg}$). The loss function is optimized and updated perturbation ($\delta\_new$) and optimized loss function ($L\_new$) is generated (Line 10). Perturbation variable ($\delta$) is updated using $\delta\_new$ (Line 11).

**Denormalizing and Normalizing Adversarial Input**: Adversarial attacks are generally performed on normalized inputs; therefore, while denormalizing the generated input, there can be information loss as the pixel values are discrete. We have noticed that those information losses can be significant in terms of energy consumption of the particular input. To make sure that the desired energy consumption of the final adversarial input can not be modified by the information loss, we denormalize and then again normalize the generated input $X'$ (Line 12 and 13). By adding this step, we ensure that there will not be any information loss while denormalizing the final adversarial input.

**Update Perturbation and Record Most Successful Perturbation**: In this step, first, we calculate number of steps induced by $X'''$ (Line

---

**Algorithm 1** Input-based Technique

```
 1: procedure INPUT-BASED TECHNIQUE(X,c)
 2:     Initialize δ, X_best, max_N
 3:     T ← number_of_iterations
 4:     iter ← 0
 5:     while iter < T do
 6:         X' ← scale(δ + X)
 7:         h_avg ← avg_step_size(X')
 8:         dist ← distance(X', X)
 9:         L ← dist + c · h_avg
10:         L_new, δ_new ← optimizer(L, δ)
11:         δ ← δ_new
12:         X'' ← denormalize(X')
13:         X''' ← normalize(X'')
14:         N ← get_no_of_steps(X''')
15:         if max_N < N then
16:             X_best ← X'''
17:             max_N ← N
18:         end if
19:     end while
20: end procedure
```

---

14). If the current number of steps in ODE solver ($N$) is greater than the recorded maximum number of iterations ($max\_N$) (Line 15), $max\_N$ variable is updated (Line 17) and the current adversarial input is recorded as highest energy-consuming adversarial sample ($X\_best$) (Line 16). When the iterations are finished, $X\_best$ is returned as the perturbed output image.

**Universal attack.** For Universal attack, we create adversarial inputs for which the energy consumption of the model during inference will be highest. In this approach, we do not preserve the semantic meaning of input. Therefore, we do not need to minimize the perturbation ($\delta$) added to the input. Hence, $\delta$ is not required to define Universal attack. Similarly as equation 1, we can define universal

attack as,

$$minimize(f(x + \delta)) \text{ such that, } (x + \delta) \in [0, 1]^n \tag{2}$$

where $x$ is the input and $f(x + \delta) = h_{avg}(x + \delta)$.

## 5 EVALUATION

We evaluate the performance of our techniques on two popular different adaptive ode solvers, **Dopri5** (Dormand and Prince (1980)) and **Adaptive Heun** (Süli and Mayers (2003)). We explore the Effectiveness and Transferability of `NODEAttack` through this section. To evaluate effectiveness of `NODEAttack`, we evaluate how much increase in energy consumption is achievable by `NODEAttack` on different datasets. To evaluate transferability of `NODEAttack`, we explore if adversarial inputs generated for one solver/architecture can increase the energy consumption for other solver/architecture.

### 5.1 EXPERIMENTAL SETUP

**Datasets and Models.** For evaluation, CIFAR-10 dataset (Krizhevsky et al. (2009)) and MNIST dataset (Deng (2012)) have been used for the training of the Neural ODE model. We use ODENet Convolutional Neural Network (CNN) models proposed by Chen et al. (2018) as the trained Neural ODE models. The main difference between model architectures used for CIFAR-10 and MNIST is in the input number of channels. For generating Input-based attack we consider c=10000 for CIFAR-10 and c=1000 for MNIST dataset.

**Baseline.** As there are no existing energy-based attack on Neural ODEs, we compare our technique with natural corruption and perturbation techniques (Hendrycks and Dietterich (2019b)) for CIFAR-10 dataset. These techniques are commonly used (Xie et al. (2020); Geirhos et al. (2018); Ovadia et al. (2019)) to evaluate the robustness of neural networks. For MNIST dataset, we use random Gaussian noise (Cattin, 2013) as baseline.

**Hardware Platform.** We use the Nvidia Jetson TX2 board for our energy consumption calculations used to evaluate Neural ODEs. To avoid noise in calculated energy consumption, we measure the energy consumption of an inference 20 times, discard the outliers and measure the mean of the remaining values.

### 5.2 EFFECTIVENESS

We have measured the effectiveness of Input-based attack by measuring the average increase in energy consumption (in Joule) during the inference with respect to benign inputs, where we measure the effectiveness of Universal attack by recording the highest energy consumption achieved by the adversarial examples during the inference. We

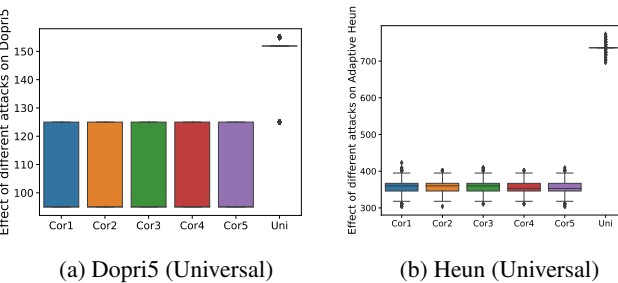

(a) Dopri5 (Universal)          (b) Heun (Universal)

Figure 3: Energy consumption induced by Universal and corruption techniques on different ODE solvers

measure the effectiveness of our approach for CIFAR-10 and MNIST datasets. For CIFAR-10 dataset, we use images generated by common perturbations and corruptions as baseline. Effectiveness of Universal attack has been measured against the corruption techniques because, in both cases, the noise present in the input is human perceptible. As the Input-based attack adds small perturbation to the input, the effectiveness of Input-based attack is measured against the perturbation techniques. For comparison, we use the five best performing corruptions and perturbations in terms of average energy consumption.

Figures 3 and 4 show the effectiveness of `NODEAttack` on Dopri5 and Adaptive Heun solvers for CIFAR-10 datasets. It can be observed that, for all four scenarios, `NODEAttack` is able to generate higher energy consuming examples than baseline methods. For Universal attack, `NODEAttack` is able to increase 125% and 50% of the energy consumption of Neural ODE model with respect to average energy consumed by benign CIFAR-10 test data, for Adaptive Heun and Dopri5 solvers, respectively. For Input-based attack, for Adaptive Heun and Dopri5 solvers respectively, `NODEAttack` is able to increase 47% and 30% of the energy consumption (in average) of Neural ODE model than energy consumed by CIFAR-10 test images.

For MNIST dataset, the average energy consumption increased by Input-based attack is 72J (31.85% avg increase) and 15J (21.4% avg increase) for Adaptive Heun and Dopri5, respectively. For the same datatset, Gaussian random noise can increase 7.9J (3.5% avg increase) and 2.1J (2.94% avg increase) for Adaptive Heun and Dopri5, respec-

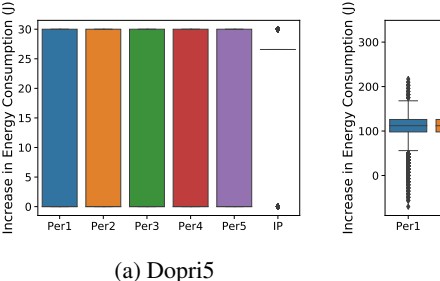
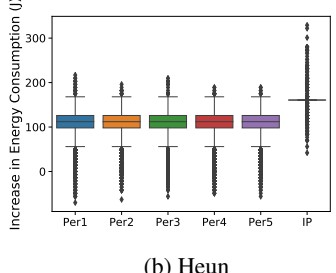

(a) Dopri5          (b) Heun

Figure 4: Energy consumption increased by Input-based and Perturbation techniques on different ODE solvers

tively. Therefore, Input-based attack outperforms the baseline in terms of increase in energy consumption. For Universal attack, the average energy consumption induced for Adaptive Heun solver is 625J (168% increase than average energy consumed by MNIST test data), where the average energy consumption induced for Dopri5 solver is 100J (42.8% increase than average energy consumed by MNIST test data).

## 5.3 TRANSFERABILITY

In this section, we try to evaluate the transferability of the Input-based attack on different ODE solvers and on network architectures. To measure the transferability, we have selected 1000 images randomly. To evaluate the transferability of our attack, we will measure two parameters: 1. What percentage of adversarial inputs can increase the energy consumption for the other solver/architecture? This is called Input Transferability Percentage (ITP) 2. What is the average percentage increase in energy consumption enforced by adversarial inputs in other solver/architecture? This is called Effectiveness Transferability Percentage (ETP). To evaluate solver-based transferability, we consider two aforementioned solver with same network architecture. Table 1 shows our findings. We can observe that transferability can exist between two solvers. It can be noticed that created adversarial inputs for Adaptive Heun are more effective against Dopri5 solver in terms of transferability.

To evaluate network-based transferability, we consider Adaptive Heun ODE solver with two different Neural ODE architecture (M1 and M2). M1 is the larger model that has an extra convolutional layer than M2. For M2 to M1 transferability, the ITP is 61.8%, however the ETP is lower (2.6%). For M1 to M2 transferability, the ITP is 80%, and the ETP is slightly higher than previously calculated ETP (4.8%). Therefore, we can observe that cross-architectural transferability is feasible for this attack. Additional evaluation on transferability can be found in the appendix.

| Type | AS \\ BS | Dopri5 | Adaptive Heun |
|------|----------|--------|---------------|
| ITP | Dopri5 | – | 85.3 |
|     | Adaptive Heun | 92.5 | – |
| ETP | Dopri5 | – | 17.09 |
|     | Adaptive Heun | 27.2 | – |

Table 1: ITP and ETP values for measuring transferability between Dopri5 and Adaptive Heun solvers. *AS* represents Attacked Solver and *BS* represents Base Solver.

## 6 CASE STUDY.

Through a case study, we have tried to demonstrate practical scenarios which can show the adverse effects of energy attack on Neural ODEs. In this section, we discuss how `NODEAttack` generated

samples can be used to design poisoning attack against Neural ODE based executable models, which are used for mobile applications. The poisoning attack is designed based on the features of popular DNN compilers; therefore, we will briefly discuss the functionality of DNN compilers first.

DNN compilers Li et al. (2020) are used to generate executable models for resource-constrained devices because Deep Learning (DL) libraries have not been able to use the hardware efficiently. DNN compilers take the model definitions described in the DL frameworks as inputs and then generate efficient code implementations on various DL hardware as outputs. The generic architecture of DNN compiler can be divided into two parts: frontend and backend. Each part uses specific intermediate data representation (IR). The frontend takes a DL model from existing DL frameworks as input Then transforms the model into the computation graph representation. DAG-based IR (in frontend) is one of the most traditional ways for the compilers to build a computation graph. In this IR, operators and tensors are treated as nodes and edges, respectively and they are organized as a directed acyclic graph (DAG). It has deficiencies such as semantic ambiguity caused by the missing definition of computation scope.

The poisoning attack is designed based on the feature of popular DNN compilers: The computational graph generated to create executable model by the DNN compiler is a Directed Acyclic Graph. The generated computational graph is created using a base input sample. When we feed a Neural ODE model to generate an executable, the generated graph assumes that the step-size calculated for the base sample is static for all samples (because the adaptive step size calculation is not captured through DAG). Because of the feasible cross-solver and cross-network transferability, if `NODEAttack` generated energy surging adversarial examples as the base sample, the energy consumption for each input will be high.

**Experiment.** In this experiment, we use two Neural ODE executable models with Adaptive Heun ODE solver to detect objects through a mobile application. Both executable models are generated based on a base CNN trained on CIFAR-10 dataset. The first executable file is initialized with a benign sample (no of iterations = 43), where the second executable file is initialized with an adversarial energy-surging sample generated through Input-based attack (no of iterations = 88). To create the executable files, we use pytorch mobile (Paszke et al. (2019)). Based on each executable model, we create an android application. For each application, we feed inputs to the model till the battery is drained fully. We also ensure that there is no other application running on the mobile during the same time.

For the executable file initialized with the benign sample, the number of classifications completed by the model is 30,188 before the battery is drained fully. While for the executable file initialized with the adversarial sample, the number of classifications done by the model is 15,102 before draining out the battery. We can notice that the adversarial examples can reduce the efficiency of the applications by approximately 50%.

## 7    CONCLUSION

In this paper, we have proposed `NODEAttack` [2] to show the vulnerability of Neural ODEs against energy-surging adversarial samples. Here, we have proposed two types of adversarial attacks: Universal attack and Input-based attack. To the best of our knowledge, we are the first to explore energy-based attack against Neural ODEs. We also observe that adversarial examples generated by `NODEAttack` can be transferable. Finally, we show the impact of `NODEAttack` generated samples on mobile application.

---

[2]https://github.com/anonymous2015258/NODEAttack

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
