# OpenReview forum: "NODEAttack: Adversarial Attack on the Energy Consumption of Neural ODEs"
_ICLR.cc/2022/Conference — ICLR 2022 Submitted_

### Official Review · Reviewer_byMA · 2021-10-16

**Correctness:** 2
**Technical Novelty And Significance:** 2
**Empirical Novelty And Significance:** 2
**Recommendation:** 3
**Confidence:** 4

**Main Review:**

My major concerns are as follows,

1. In Section 5.1, $E_i$ and $E_u$ are not clear.
a) The authors mention that 'We want to add perturbation $\delta$ to x such that energy consumption is maximum'. However, why does the formula of $E_i$ and $E_u$ contain -max? They are very strange.
b) On the other hand, Why $E_u$ does not contain perturbation $\delta$?
c) The authors mention 'By increasing the value of $E_i$ and $E_u$, energy robustness of a model can be increased.'. Why this statement holds?

2. The paper writing is not clear and some parts of the paper are hard to understand. For example,
a) At the end of Section 4, the authors use '15J' to demonstrate the energy consumption. J is not clear. From my perspective, J should be Joule. Please make it clear.
b) In Eq.(1), it is not clear which parameter needs to be minimized.
c) In Eq.(1), What is c? Why do we need to use c?

3. In Figure 1, why the average energy consumption is a line for the 'iters=3' case?

4. In the 'Update Perturbation and Record Most Successful Perturbation' subsection of Section 5.3,  the sentence 'greater than the recorded minimum loss (max_N)' is not clear.

5. In Eq.(2), why the minimization is only with respect to $f(x+\delta)$? I understand this is for the universal attack. However, the authors should mention what is the significant difference between Eq.(1) and Eq.(2). Otherwise, Eq.(2) is very hard to understand.

6. In the Baseline part of Section 6.1, the authors mention CIFAR-10-C and CIFAR-10-P. However, both of them should be in the part of the dataset instead of this Baseline part. Therefore, it is not clear what is the exact baseline methods. In addition, the authors mention two methods: corruption and perturbation techniques. Given only two methods' name is not enough. The authors should briefly introduce these two methods.

7. At the end of Section 6.2, the authors discuss the experiments on the MNIST datasets. But why the authors do not provide figures to show the performance like Figures 3 and 4? This experiment setting is not clear enough.

8. In Section 6.3, the authors mention 'we will measure two parameters: '. However, I think ITP and ETP both are evaluation metrics instead of parameters. And the authors should provide references for these two metrics and demonstrate they are widely used in the existing works.

9. In Section 6.3, the authors mention 'the larger model has an extra convolutional layer than smaller mode'. However, it is not clear which model is large and which model is small.

10. In Section 7. Why the no. of iterations on the benign sample and input-based attack are different? Since the number of iterations is different, it is hard to say the compared performance results. The comparison is not fair.

11. In Algorithm 1, lines 12-18, Why X-best is based on the condition of max_N<N? According to Eq.(1), the authors would like to minimize $\delta$. From my perspective, if the attack is successful and N<max_N, the perturbation should be updated and saved. Therefore, I believe this part of Algorithm 1 is not correct.


**Summary Of The Paper:**

In this paper, the authors proposed an attack model named NODEAttack to verify the vulnerability of Neural ODEs against energy-surging adversarial samples. The authors study two attack cases: Input-based attack (untargeted attack) and universal untargeted attack. The experiments on CIFAR-10 and MNIST datasets show the reasonable performance of the proposed model outperforms common perturbations and corruptions methods. A case study based on DNN compilers and PyTorch mobile demonstrates the generated adversarial examples can reduce the efficiency of real-world applications.

**Summary Of The Review:**

In general, I feel this paper has several limitations on the technique. Some technical details are not correct. The paper presentation is not clear. Some experiment settings are not reasonable.

I suggest reject this paper.

---

> ### Author Response · Authors · 2021-11-23
> **Author Response**
>
> Thank you for your feedback on our work. Here are our comments  for the asked queries.
> First we apologize for the confusion caused by our writing in some places. While we have updated the paper to avoid these misunderstandings, we also try to explain a few points which have created confusion.
> 1. ETP and ITP: These metrics have been proposed by us and have not been used earlier. We have discussed in the appendix (Section G in appendix) about the usage of these two metrics.
> 2. CIFAR-10-P and CIFAR-10-C: These datasets are prepared by adding perturbation and corruption techniques to CIFAR-10 dataset. We use those perturbation and corruption techniques to evaluate NODEAttack.
> 3. Different evaluation for MNIST and CIFAR-10: As MNIST dataset is 1D, all the perturbation and corruption techniques can not be applied to the dataset. We have used random gaussian noise as a baseline against the MNIST dataset.
> 4. No of iterations in Section 7: The number of iterations mentioned in the case study represents the number of iterations needed to approximate the function for that input. This implies the energy consumed by the particular input. As we wanted to evaluate the effect of NODEAttack on a real time device, we selected a benign (low NOI) and an adversarial input (high NOI).
> 5. Max_N and N in algorithm: Max_N represents the maximum number of iterations achieved by the perturbed input till an iteration. If N is greater than max_N, that means we have found an input that can achieve a higher number of iterations than the  number of iterations achieved by other inputs till that point. Hence, we update the value of max_N. Therefore, we think that our algorithm is correct.
> 6. Ei and Eu: We have added additional explanations for Ei and Eu. Also, delta is used to add perturbations to the benign inputs such that the perturbed input remains within the distribution. As Eu considers both In-distribution and Out-of-distribution inputs, delta is not needed to define Eu.
> 7. Figure 1 Explanation: ‘iters=3’ represents one line because very few benign test inputs use 3 iterations for approximation and variance of energy consumption is low. Therefore, the boxplot only shows a line.

---

### Official Review · Reviewer_v5cj · 2021-11-02

**Correctness:** 3
**Technical Novelty And Significance:** 3
**Empirical Novelty And Significance:** 2
**Recommendation:** 6
**Confidence:** 4

**Main Review:**

Instead of common performance/accuracy robustness, this paper considered the uncertainty of energy consumption of neural ODE models. This is very interesting and inspires the community to move eyes on the robustness of ML algorithms in terms of other metrics besides the performance.

This paper proposed two notions of energy robustness, the input-based energy robustness $E_i$ and the universal energy robustness $E_u$m. It used an intuitive metric, the average step_size $h_{avg}$, to measure the energy consumption.

This paper focuses on two adaptive ODE solvers, Dopri5 and Heun. Experimental results on MNIST and CIFAR10 demonstrate the effectiveness and transferability of the proposed attack.

Some questions:
1. The imperceptibility: The generated data in the appendix seem very noisy and easy to distinguish from clean data. How well does the attack perform if the budget is limited to be small?
2. The adversarial data are crafted for the energy-consumption attacks. How much do they affect the test accuracy?
3. Can the authors discuss a bit on the defense of these adversarial examples?

**Summary Of The Paper:**

This paper considered the robustness of energy consumption of neural ODEs and proposed NODEAttack to generate adversarial inputs that maliciously increase the computation overhead of NODE solvers. Experimental results verify the effectiveness of the proposed NODEAttack and the transferability among different ODE solvers.

**Summary Of The Review:**

I will update this part after the review phase...

---

> ### Author Response · Authors · 2021-11-23
> **Author Response**
>
> Thank you for your positive feedback on our work. Here are our comments  for the asked queries.
> 1. Imperceptibility: We have added a section in the appendix (Section F in appendix) where we compare percentage increase of energy consumption with respect to Input-based attack scenario by decreasing the c value significantly (high c represents high perturbation).
> 2. Feasible Defense: We discuss feasible defense mechanisms for two attack mechanisms in the updated appendix (Section D in appendix).
> 3. Accuracy Robustness vs Energy Robustness: To understand the difference between accuracy robustness and energy robustness, we randomly select 500 CIFAR-10 and MNIST images and apply Input-based attack on those inputs. We have added the experimental results in the appendix (Section E in appendix).  While we can not conclude that  accuracy robustness and energy robustness of Neural ODEs are related, we can notice that accuracy robustness of the Neural ODE model trained with CIFAR-10 dataset is higher than model trained with MNIST dataset.

---

### Official Review · Reviewer_BU3E · 2021-11-02

**Correctness:** 3
**Technical Novelty And Significance:** 3
**Empirical Novelty And Significance:** 3
**Recommendation:** 6
**Confidence:** 4

**Main Review:**

Strengths

+ The paper adds to the growing literature on neuralODEs which have gained more traction in resource-constrained devices due to their memory efficiency. It raises a very valid concern about the vulnerability of these models to energy attacks. The reviewer finds this to be the main contribution of the paper (based on the assumption that this is the first paper to do so).

+ The paper defines energy robustness for Neural ODEs in two ways: Input-based Energy Robustness and Universal Energy Robustness (Eu). The first is defined based on the maximum energy consumed by the model for an input which belongs to the training data distribution of the model. The second can be described based on the highest possible energy consumed by a model for any input.

+ The paper also evaluates the transferability of NODEAttack, that is, if adversarial inputs are generated for one solver/architecture can they also increase the energy consumption for other solver/architecture.

Weaknesses/Issues that would make the paper stronger :

- What would be the tradeoff between robustness and performance if the adaptive step-sizing was not used?

- Could you add a discussion on how the presented observations and results on vulnerability of ODEs to energy attacks compare with those on other forms of adaptive inference?

**Summary Of The Paper:**

Neural ODE (Ordinary Differential Equation) models  have emerged as an effective architecture due to their low memory usage which makes them attractive for resource-constrained devices. Further, the inference in these models is adaptive. This paper investigates whether this adaptive inference can be used by an attacker to launch an energy attack on the neural ODEs. While energy attacks on adaptive inference has been studied before, this paper is the first paper (to the best of the knowledge of the reviewer) to study this on neural ODEs. This study is well-motivated, and the experimental evaluation is convincing (though it is far from complete in terms of the used attack methods or configurations of the used methods). 2 solvers and 2 datasets are used in the experiments. Overall, the paper definitely adds to the state of the art on neural ODEs and robust learning.

**Summary Of The Review:**

The paper makes a clearly identifiable contribution to the literature by demonstrating energy attacks on neural ODEs. Hence, the reviewer leans towards accepting this paper. A more thorough analysis would have made the paper stronger and more influential.

---

> ### Author Response · Authors · 2021-11-23
> **Author Response**
>
> Thank you for your positive feedback on our work. Here are our comments  for the asked queries.
> 1. Trade-off between adaptive and fixed step size: As a feasible defense mechanism, we discuss sing fixed number of iterations (i.e., fixed step size) in Neural ODEs in the updated appendix (Section D.2 in appendix). We explain how fixed step-size can negatively impact the approximation of the ODE solver, therefore decreasing accuracy-robustness.
> 2. Comparison Between ODEs and other AdNNs: We have added an additional section in the appendix comparing ODEs and other AdNNs (Section I in appendix).

---

> > ### Comment · Reviewer_BU3E · 2021-11-30
> > **Thank you**
> >
> > I thank the reviewer for doing the requested comparison and adding it to the appendix.

---

### Official Review · Reviewer_BRCY · 2021-11-05

**Correctness:** 3
**Technical Novelty And Significance:** 2
**Empirical Novelty And Significance:** 2
**Recommendation:** 3
**Confidence:** 4

**Main Review:**

### Strengths:
- New energy consumption attack against Neural ODE models (previous works consider energy consumption attacks against Adaptive Neural Networks).

### Weaknesses:
- The paper is poorly organized and can be improved, e.g. 'Background' and 'Related works' sections can be merged together. Section 4 can be expanded to include more background information about Neural ODE.
- Description of the equations can be improved. Some equations have errors, e.g. eq. 1 on p. 5 minimize the regularized loss: δ + c × f(x + δ), when it should be: |δ| + c × f(x + δ).
- The optimal attack is selected only to increase the number of ODE iterations (line 15-17 in Algorithm 1). Trade-offs between the attack success rate and the attack energy consumption can be considered.
- The authors didn't consider a simple detection strategy based on the number of ODEs iterations on benign examples. It will be interesting to see if the proposed attack can be circumvented using this simple defense.
- Input-based Energy Robustness Attack is not very successful and doesn't increase energy consumption significantly. Universal Energy Robustness Attack is more successful but might produce out-of-distribution examples, which might be easily detected. The authors should perform additional analysis to consider the detection of adversarial examples generated by the Universal Energy Robustness Attack.


**Summary Of The Paper:**

The authors propose an adversarial test-time attack on Neural ODE models, which increases the inference time of the NODE models. Two variants of the attack are introduced. The proposed attacks are evaluated on CIFAR10 and MNIST datasets. In experiments, the authors showed the proposed attacks increased the inference time of NODE models for object detection task, which drained the battery faster on mobile device.


**Summary Of The Review:**

The authors introduced an energy consumption attack on Neural ODE models, which is a novel application of energy consumption attacks. This is a novel application idea, but the idea is not technically novel. Furthermore, the paper is poorly organized, and experimental analysis can be improved. For example, the authors can add the analysis with simple defense strategies such as thresholding the number of ODEs iterations or out-of-distribution detection of adversarial examples generated by universal energy attack.

---

> ### Author Response · Authors · 2021-11-23
> **Author Response**
>
> Thank you for your feedback on our work. Here are our comments  for the asked queries.
> 1. Writing errors have been addressed in the new draft.
> 2. Trade-off between Energy Consumptions: We acknowledge the fact that energy consumption increased through NODEAttack for specific inputs is lesser than the energy consumption required to generate those inputs (as NODEAttack is an iterative process). However, as mentioned in the case study, a single adversarial input can slow down an application significantly. Also, because of the feasible attack transferability, the poisoning attack mentioned in case study becomes a considerable threat.
>  3. Feasible Defense: We discuss feasible defense mechanisms for two attack mechanisms in the updated appendix (Section D in appendix). While we can empirically find that Universal attack can be defended using low energy consuming binary classifier, using fixed number of iterations (i.e., fixed step size) can negatively impact the approximation of the ODE solver.

---

### Author Response · Authors · 2021-11-23
**General Response**

In the rebuttal, we have added the following components to our draft.
1. Discussion about feasible defense (Section D in appendix)
2. Additional baseline for MNIST dataset (Section 5.1 and 5.2 in paper)
3. Imperceptibility Study (Section F in appendix)
4. A comparison between Accuracy Robustness and Energy Robustness (Section E in appendix)
5. Comparison between AdNNs and Neural ODE (Section I in appendix)
6. Brief discussion about baseline techniques  (Section H in appendix)

---

### Decision · Program_Chairs · 2022-01-20

**Decision:**

Reject

**Comment:**

The paper proposes an energy consumption attack to neural ODE models. There are two complains from the reviewers:
- Although this is a new application to energy consumption attack, most of the attack techniques are simple extensions to the previous attack papers, so the novelty is questioned by some of the reviewers.
- The paper is poorly written.
We therefore decide to reject the paper and encourage the authors to address the concerns in their next submission. Reviewers also think a careful discussion about the defense or detection mechanism against the proposed attack will be a good thing to add.